# Cervical cancer screening improvements with self-sampling during the COVID-19 pandemic

**Miriam Elfström[1,2], Penelope Grace Gray[1], Joakim Dillner[1]\***

[1]Center for Cervical Cancer Elimination, F46, Pathology and Cancer Diagnostics, Medical Diagnostics Karolinska, Karolinska University Hospital and Division of Cervical Cancer Elimination, CLINTEC, Karolinska Institutet, Stockholm, Sweden; [2]Regional Cancer Center of Stockholm-Gotland, Cancer Screening Unit, Sweden, Stockholm, Sweden

## Abstract

**Background:** At the onset of the COVID-19 pandemic cervical screening in the capital region of Sweden was canceled for several months. A series of measures to preserve and improve the cervical screening under the circumstances were instituted, including a switch to screening with HPV self-sampling to enable screening in compliance with social distancing recommendations.

**Methods:** We describe the major changes implemented, which were (1) nationwide implementation of HPV screening, (2) switch to primary self-sampling instead of clinician sampling, (3) implementation of HPV screening in all screening ages, and (4) combined HPV vaccination and HPV screening in the cervical screening program.

**Results:** A temporary government regulation allowed primary self-sampling with HPV screening in all ages. In the Stockholm region, 330,000 self-sampling kits were sent to the home address of screening-eligible women, instead of an invitation to clinician sampling. An increase in organized population test coverage was seen (from 54% to 60% in just 1 year). In addition, a national campaign for faster elimination of cervical cancer with concomitant screening and vaccination for women in ages 23–28 was launched.

**Conclusions:** The COVID-19 pandemic necessitated major changes in the cervical cancer preventive strategies, where it can already be concluded that the strategy with organized primary self-sampling for HPV has resulted in a major improvement of population test coverage.

**Funding:** Funded by the Swedish Association of Local Authorities and Regions, the Swedish Cancer Society, the European Union's Horizon 2020 Research and Innovation Program, the Swedish government, and the Stockholm county.

**\*For correspondence:** Joakim.Dillner@ki.se

**Competing interest:** The authors declare that no competing interests exist.

## Editor's evaluation

This is a valuable piece of work given the scope of the intervention(s) and the precedent it sets i.e. a crisis can in fact accelerate positive changes in screening that have been academic possibilities rather than practical realities. The evidence is solid since data were obtained from the national cervical screening registry during the pandemic. The work will be of broad interest to researchers and policy makers involved in cervical cancer screening.

## Introduction

Prior to the COVID-19 pandemic outbreak in spring 2020, the Swedish cervical cancer prevention efforts consisted of invitations for screening at maternity clinics at an interval of 3–7 years. The National Board of Health and Welfare decided in 2015 that screening of 30- to 70-year-old women should primarily be performed with an HPV test and screening of 23- to 29-year-old women with cytology (*National Board of Health and Welfare, 2015*). However, 5 years later when surveyed during the autumn of the pandemic, 5/21 regions in Sweden had still not implemented the national program. Self-sampling targeting long-term non-attenders as a method to increase population coverage was recommended but was rarely used (*Swedish National Cervical Screening Registry, 2021*). There was school-based vaccination of both girls and boys (with high population coverage) (*National Board of Health and Welfare, 2022*) but no consideration of strategies for an even faster strategy to eliminate cervical cancer by concomitant screening and vaccination of young women (*Dillner et al., 2021*). Although HPV testing has been shown to have higher sensitivity in all age groups and higher specificity for women aged 30 or older compared to cytology-based screening, it was previously not recommended below 30 years of age largely because of the high prevalence of HPV infections, most of which will be cleared without causing cellular lesions or cancer, leading to overdiagnosis and over-treatment (*Arbyn et al., 2012*; *Leinonen et al., 2009*; *Ronco et al., 2014*). However, an increasing proportion of young women that are entering the screening program have been vaccinated, resulting in lower HPV prevalences and lower predictive values of screening (*Lei et al., 2020b*). Cellular abnormalities among young, vaccinated women are still seen, but typically contain only non-progressive HPV types (*Kann et al., 2020*).

In April of 2020 all non-emergency healthcare was stopped in the capital region of Sweden because of a severe COVID-19 outbreak. Consequently, 192,000 cervical cancer screening invitations with appointments were canceled. In June the same year, the screening program was allowed to restart but the midwife clinics could not be used for screening as usual, to avoid crowding. The Swedish National Board of Health and Welfare enacted a temporary regulation that (1) allowed for primary self-sampling instead of clinician-based sampling and (2) allowed for primary HPV-based screening in all ages between 23 and 70 years of age. It is well known that when self-samples are analyzed using an HPV assay based on polymerase chain reaction, the sensitivity is not inferior to samples taken by medically trained staff (*Arbyn et al., 2018*) and that self-sampling can be used to increase population coverage of screening, with improved attendance among under-screened and hard-to-reach women (*Elfström et al., 2019*; *Winer et al., 2019*).

## Methods

### Study population and data collection

Most data in this report derive from publicly available data on new Swedish regulations and strategies used during the pandemic (major official websites https://www.socialstyrelsen.se and https://www.regeringen.se). Results on population coverage and number of screening invitations/self-sampling kits sent are derived from the website of the Swedish National Cervical Screening Registry of Sweden (https://www.nkcx.se/). The registry collects all data on cervical screenings and invitations in Sweden and compares the data to aggregated data from the population registry to calculate the population test coverage of screening (*Elfström et al., 2016*). To evaluate the contribution of clinician taken samples or self-samples on the population test coverage of the organized screening program, the population coverage was calculated stratified by mode of sampling (self-sampled or clinician sampled) by year, age group, and calendar year.

The registry linkages using the NKCx were approved by the National Ethical Review Agency of Sweden (decision number 2023-00289-02). The agency is appointed directly by the government of Sweden, chaired by a senior judge and has the authority to determine requirements for consent, was not required for this study. Key factors in the ethical consideration include whether the integrity risks were minimal (strong security measures and limited access to data) and benefits and disadvantages of the research. In this case, population-based estimates were sought, which by definition means that the entire population is studied.

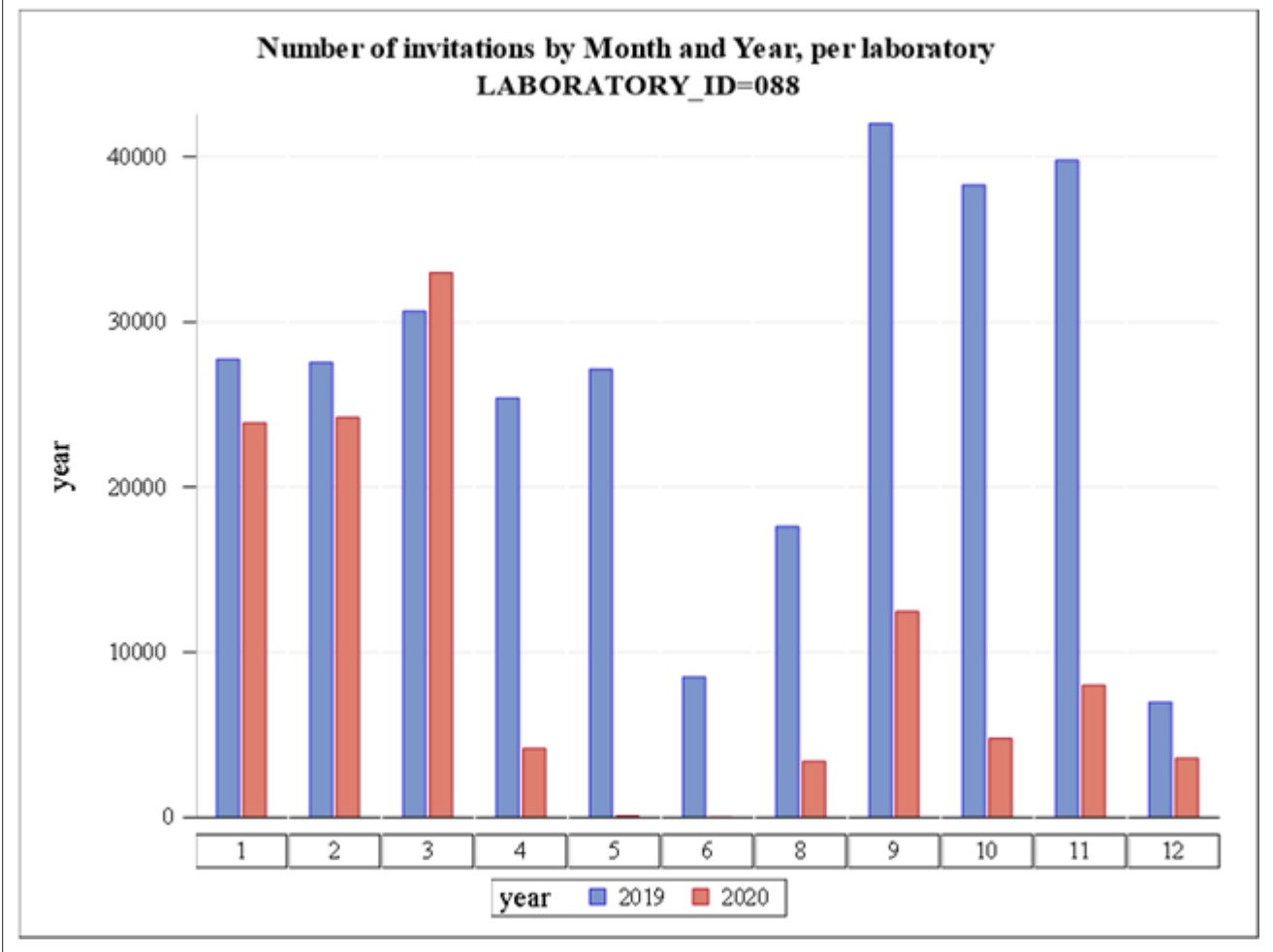

**Figure 1.** Number of invitations to cervical cancer screening in Stockholm by month and year. LabID 088 is Karolinska University Hospital (Stockholm region). Numbers on the *X*-axis refer to calendar month.

## Results

The number of cervical screening invitations sent per month in the Stockholm region during 2019 and 2020 is shown in *Figure 1*. The formal decision to cancel all screening was effective April–June 2020, during which time there was a severe COVID-19 outbreak in the Stockholm region (*Pimenoff et al., 2021*). When screening was re-initiated in the second half of 2020, only a limited number of invitations were allowed per screening station in order to avoid crowding (*Figure 1*). During the second half of 2020, only women invited for their first cervical screen and women due for follow-up after abnormal screening results were invited to clinician sampling. Clinician-based sampling was also recommended for women who were pregnant when they received their invitation to self-sampling. There were preparations for a switch to primary self-sampling as the prime screening modality, which was launched in March 2021. Except for women at first cervical screen, pregnant women and follow-up after abnormal screens, self-sampling kits were mailed to all eligible women in the population (direct send instead of invitation to maternity care clinic – no requirement of having to order the kit). A detailed description of the self-sampling kits and their use can be found in *Elfström et al., 2019*.

The organized screening program is based on first generating a list of eligible women who (1) are resident in the catchment area (in this case the greater Stockholm region, with about 2 million resident inhabitants), (2) did not take a cervical test during the recommended age-specific screening interval (3 years 23–49 years of age, 7 years 50–70 years of age). This is assessed by importing of files with tests performed from all laboratories (both public and private) in the region and comparing the sampling dates and the personal identification numbers with the population registry of resident

**Table 1.** Test coverage of organized cervical screening (% of total population tested) in the Stockholm and Gotland region during 2013–2021 stratified by age group.

The target age group of the program was 23–60 years of age until 2015 and 23–70 years of age from 2015 onwards. Program cancellation due to the COVID-19 pandemic occurred in April 2020 and the switch to primary screening with self-sampling was implemented in March 2021.

| | Test coverage (%) | | | | | | |
|---|---|---|---|---|---|---|---|
| | Age group (years) | | | | | | |
| Year | 23–70 | 23–25 | 26–30 | 31–40 | 41–50 | 51–60 | 61–70 |
| 2013 | 47.7 | 74.7 | 60.0 | 58.6 | 56.9 | 49.3 | 0.25 |
| 2014 | 48.7 | 77.2 | 59.7 | 59.5 | 57.6 | 51.5 | 0.27 |
| 2015 | 51.1 | 77.3 | 59.8 | 59.2 | 58.8 | 53.9 | 10.3 |
| 2016 | 54.0 | 78.2 | 62.5 | 61.0 | 61.0 | 57.7 | 14.7 |
| 2017 | 56.2 | 80.6 | 64.2 | 62.3 | 62.4 | 61.6 | 17.4 |
| 2018 | 58.2 | 83.6 | 66.5 | 63.1 | 64.2 | 63.1 | 20.3 |
| 2019 | 60.5 | 85.4 | 68.2 | 64.6 | 65.9 | 62.7 | 27.9 |
| 2020 | 54.3 | 81.7 | 62.3 | 56.7 | 57.2 | 57.6 | 25.4 |
| 2021 | 59.6 | 88.9 | 69.6 | 62.3 | 62.2 | 59.6 | 32.6 |

women (3) checking that they did not opt out of the screening program. This is not common, but out of the 732,276 resident women in the eligible target ages, 2655 women (0.4%) had opted out of the program. In the switch that was implemented in March 2021, the previous policy of sending an invitation letter with an appointment time for cervical screening at a local Maternity Care clinic was for most women (see above) replaced by sending of a self-sampling kit (including instructions and a pre-paid return envelope). During 2021, approximately 320,000 self-sampling kits were mailed. There was an increase in the population coverage of the organized cervical screening program, from 54% to 60% (**Table 1**). The improvement was seen in most age groups and the population test coverage levels approached the test coverage levels in the year before the COVID-19 outbreak when the program was canceled (**Table 1**). There has been a steady increase in population test coverage over the past decade (**Figure 2**). The trend was broken during the pandemic, but the trend seems to be restored in 2021 (**Figure 2**).

Because population test coverage is calculated over the length of a full screening interval, most of the population test coverage was still attributable to tests taken already before the pandemic. As shown in **Table 2**, about two thirds of the coverage was still attributable to clinician-based sampling

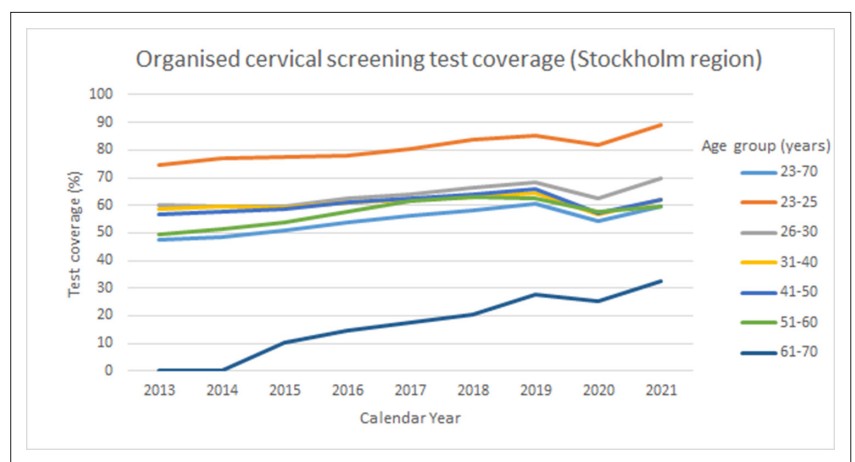

**Figure 2.** Test coverage of organized cervical screening among women in the Stockholm Gotland region, Sweden between the years 2013 and 2021.

**Table 2.** Test coverage of organized cervical screening stratified by mode of index sampling (clinician taken samples or self-sampling) in the Stockholm Gotland region during 2013–2021 stratified by age group.

| | Absolute test coverage (%) from index samples taken by clinician | | | | | | |
|---|---|---|---|---|---|---|---|
| | Age class | | | | | | |
| Year | 23–70 | 23–25 | 26–30 | 31–40 | 41–50 | 51–60 | 61–70 |
| 2013 | 47.7 | 74.7 | 60.0 | 58.6 | 56.9 | 49.3 | 0.25 |
| 2014 | 48.7 | 77.2 | 59.7 | 59.5 | 57.6 | 51.5 | 0.27 |
| 2015 | 51.1 | 77.3 | 59.8 | 59.2 | 58.8 | 53.9 | 10.3 |
| 2016 | 54.0 | 78.2 | 62.5 | 61.0 | 61.0 | 57.7 | 14.7 |
| 2017 | 56.2 | 80.6 | 64.2 | 62.3 | 62.4 | 61.6 | 17.4 |
| 2018 | 58.2 | 83.6 | 66.5 | 63.1 | 64.2 | 63.1 | 20.3 |
| 2019 | 60.0 | 85.4 | 68.2 | 64.1 | 65.3 | 61.9 | 27.5 |
| 2020 | 53.1 | 81.7 | 62.3 | 55.7 | 55.5 | 55.4 | 24.2 |
| 2021 | 41.0 | 86.8 | 46.2 | 41.3 | 39.5 | 42.7 | 19.8 |
| | Absolute test coverage (%) from index sample taken by self-sampling | | | | | | |
| | Age class | | | | | | |
| Year | 23–70 | 23–25 | 26–30 | 31–40 | 41–50 | 51–60 | 61–70 |
| 2013 | 0.00 | 0.00 | 0.00 | 0.00 | 0.00 | 0.00 | 0.00 |
| 2014 | 0.00 | 0.00 | 0.00 | 0.00 | 0.00 | 0.00 | 0.00 |
| 2015 | 0.00 | 0.00 | 0.00 | 0.00 | 0.00 | 0.00 | 0.00 |
| 2016 | 0.00 | 0.00 | 0.00 | 0.00 | 0.00 | 0.00 | 0.00 |
| 2017 | 0.00 | 0.00 | 0.00 | 0.00 | 0.00 | 0.00 | 0.00 |
| 2018 | 0.00 | 0.00 | 0.00 | 0.00 | 0.00 | 0.00 | 0.00 |
| 2019 | 0.46 | 0.00 | 0.00 | 0.49 | 0.60 | 0.78 | 0.37 |
| 2020 | 1.22 | 0.00 | 0.00 | 0.48 | 1.64 | 2.26 | 1.16 |
| 2021 | 18.6 | 2.17 | 23.4 | 21.0 | 22.7 | 16.9 | 12.8 |

also in the age groups 26–70 where there had been no possibility for organized screening using clinician taken samples (except for pregnant women) since the start of the pandemic.

As Sweden uses an annual call & recall system, a certain proportion of the population test coverage is attributable to participation by previous non-attenders (*Table 3*). The proportion of the test coverage that was attributable to long-term non-attenders taking self-samples was 3.9%, a large part of the overall increase in population test coverage seen (*Table 3*). Note that self-samples were not used at all before 2019 and were used only in a small scale project targeting non-attenders in 2019–2020 (*Table 3*).

The increased attendance using self-sampling among prior non-attenders is particularly evident when population coverage is expressed as a proportion of the sampling mode (*Table 4*). Whereas the proportion of the test coverage attributable to clinician sampling of prior non-attenders has always been less than 10% of the samples, >20% of the population test coverage attributable to self-sampling was attributable to prior non-attenders (*Table 4*).

As self-sampling cannot be used for cytology, only for HPV testing, the need to use self-sampling promoted a change to more widespread use of HPV testing. Although HPV testing had been mandated from age 30 and upwards already in 2015, in 2020 there were still 5 counties that did not use it. This changed and since autumn 2021 all counties in Sweden now use primary HPV screening. In the age groups below 30, HPV screening had not previously been recommended because of concerns about over-screening in an age where HPV infections were common. However, HPV prevalences

**Table 3.** Absolute test coverage (%) of organized cervical screening due to attendance of prior long-time non-attenders stratified by mode of index sampling (clinician taken samples or self-sampling) in the Stockholm Gotland region during 2013–2021 stratified by age group.

| | **Absolute test coverage (%) attributable to previous long-time non-attenders* giving clinician taken samples** | | | | | |
|---|---|---|---|---|---|---|
| | **Age class** | | | | | |
| Year | **23–70** | 23–25 | 26–30 | 31–40 | 41–50 | 51–60 | 61–70 |
| 2013 | **4.09** | Na | 3.01 | 8.79 | 4.45 | 3.47 | 0.02 |
| 2014 | **4.26** | Na | 3.08 | 9.21 | 4.74 | 3.50 | 0.03 |
| 2015 | **4.54** | Na | 3.04 | 9.58 | 5.09 | 3.73 | 0.57 |
| 2016 | **4.95** | Na | 3.01 | 10.2 | 5.62 | 4.19 | 0.93 |
| 2017 | **5.30** | Na | 3.31 | 10.7 | 5.89 | 4.60 | 1.20 |
| 2018 | **5.71** | Na | 3.81 | 11.3 | 6.15 | 5.01 | 1.49 |
| 2019 | **5.76** | Na | 3.83 | 11.5 | 5.89 | 4.91 | 1.81 |
| 2020 | **5.11** | Na | 3.61 | 10.2 | 4.96 | 4.30 | 1.49 |
| 2021 | **3.57** | Na | 2.40 | 7.14 | 3.19 | 3.24 | 1.13 |
| | **Absolute test coverage (%) attributable to previous long-time non-attenders* giving self-samples** | | | | | |
| | **Age class** | | | | | |
| Year | **23–70** | 23–25 | 26–30 | 31–40 | 41–50 | 51–60 | 61–70 |
| 2013 | **0.00** | Na | 0.00 | 0.00 | 0.00 | 0.00 | 0.00 |
| 2014 | **0.00** | Na | 0.00 | 0.00 | 0.00 | 0.00 | 0.00 |
| 2015 | **0.00** | Na | 0.00 | 0.00 | 0.00 | 0.00 | 0.00 |
| 2016 | **0.00** | Na | 0.00 | 0.00 | 0.00 | 0.00 | 0.00 |
| 2017 | **0.00** | Na | 0.00 | 0.00 | 0.00 | 0.00 | 0.00 |
| 2018 | **0.00** | Na | 0.00 | 0.00 | 0.00 | 0.00 | 0.00 |
| 2019 | **0.46** | Na | 0.00 | 0.49 | 0.59 | 0.77 | 0.00 |
| 2020 | **1.21** | Na | 0.00 | 0.48 | 1.62 | 2.24 | 1.14 |
| 2021 | **3.86** | Na | 1.53 | 5.83 | 4.55 | 4.08 | 2.70 |

*Long-time non-attenders are defined as women who have no history of any screening samples (neither by organized nor disorganized screening) prior to the index organized screening sample for 7.5 years (for those aged 23–49 years old) or 9.5 years (for those aged 50–70 years old).

are dropping because of vaccination and the emergency interim guidelines had now allowed HPV screening in all ages.

After the pandemic, permanent regulations were issued that came into effect on 2022-07-01. These allow choice between primary self-sampling and sampling by a clinician and have also changed the age for HPV screening to be mandated between 23 and 70 years of age. In other words, the changes that were required because of the pandemic have resulted in that huge and permanent improvements could be implemented.

## Launch of an even faster cervical cancer elimination campaign

The EVEN FASTER concept for rapid cervical cancer elimination is based on concomitant HPV vaccination and HPV screening targeting the age groups where the HPV infection is spread (have an effective reproductive number >1) (*Dillner et al., 2021*). HPV vaccination without concomitant testing is most effective among subjects before sexual debut, as they are HPV negative and the vaccine does not prevent infections that have already occurred (*Lei et al., 2020a*). However, among women after sexual debut who test HPV negative, the vaccine is equally effective as among subjects before sexual debut (*Apter et al., 2015*). By concomitant vaccination and HPV screening, HPV negative women will

**Table 4.** Proportion (%) of the test coverage attributable to previous long-time non-attenders stratified by mode of index sampling (clinician taken samples or self-sampling) in the Stockholm Gotland region during 2013–2021 stratified by age group.

| | Proportion (%) of test coverage (clinician taken index samples) attributable to previous long-time non-attenders* | | | | | | |
|---|---|---|---|---|---|---|---|
| | **Age class** | | | | | | |
| Year | **23–70** | 23–25 | 26–30 | 31–40 | 41–50 | 51–60 | 61–70 |
| 2013 | **8.59** | na | 5.01 | 15.0 | 7.82 | 7.03 | 10.0 |
| 2014 | **8.76** | na | 5.15 | 15.5 | 8.23 | 6.81 | 9.63 |
| 2015 | **8.88** | na | 5.08 | 16.2 | 8.65 | 6.92 | 5,52 |
| 2016 | **9.17** | na | 4.83 | 16.7 | 9.21 | 7.25 | 6.31 |
| 2017 | **9.42** | na | 5.16 | 17.2 | 9.44 | 7.48 | 6.93 |
| 2018 | **9.81** | na | 5.73 | 17.9 | 9.58 | 7.94 | 7.36 |
| 2019 | **9.59** | na | 5.62 | 17.9 | 9.02 | 7.94 | 6.56 |
| 2020 | **9.62** | na | 5.80 | 18.4 | 8.93 | 7.77 | 6.17 |
| 2021 | **8.72** | na | 5.19 | 17.3 | 8.08 | 7.58 | 5.70 |
| | Proportion (%) of test coverage (self-sampled index samples) attributable to previous long-time non-attenders* | | | | | | |
| | **Age class** | | | | | | |
| Year | **23–70** | 23–25 | 26–30 | 31–40 | 41–50 | 51–60 | 61–70 |
| 2021 | **20.7** | na | 6.54 | 27.8 | 20.0 | 24.1 | 21.0 |

*Long-time non-attenders are defined as women who have no history of any screening samples (including via both organized and disorganized screening) prior to the index organiszed screening sample for 7.5 years (for those aged 23–49 years old) or 9.5 years (for those aged 50–70 years old).

reap the full benefit of protection by vaccination and the HPV-positive women can be followed up as usual in the screening program (and are thus also protected against cervical cancer). Although the first version of the concept of concomitant screening and vaccination was published 7 years ago (*Bosch et al., 2016*), there had been no consideration of whether to actually implement it.

After the acute phase of the epidemic, the setting changed. There was a large screening deficit and multiple strategies were needed in order to ensure that it could be managed without adverse effects for the women. Also, very effective mass vaccination campaigns against COVID-19 had been successfully launched and the switch to self-sampling resulted in that Maternity Care Clinics were interested in advancing their offer for in-person visits by providing both vaccination and screening to women coming for their first screening visit.

It is expected that if there is a strong reduction in the circulation of cancer-causing HPV types, the screening and follow-up efforts can be concentrated to those few women who still test positive for oncogenic HPV types thus greatly improving the ability to cope with the screening deficit and enabling Sweden to faster reach the WHO target of elimination of cervical cancer.

## Discussion

A large COVID-19 outbreak in the Stockholm region necessitated concentration of healthcare to emergency care and cancer screening was canceled. We describe that the major measures taken to mitigate the screening deficit (organized primary self-sampling and an even faster campaign with concomitant screening and vaccination) resulted in several important and lasting improvements of the cervical cancer prevention program in Sweden.

The roll-out of primary self-sampling was done in the context of routine screening which means that decisions were made consecutively and the program was adapted as needed to meet the changing dynamics of the pandemic. Therefore, this analysis of the response was completed post hoc. Given that Sweden has a comprehensive registry of all invitations (including mailing of self-sampling materials) and all cervical tests performed in the country, it was possible to perform a detailed description and evaluation of the switch retrospectively. The national even faster campaign with concomitant

screening and vaccination was launched as a formal Phase IV trial with the protocol registered at clinicaltrials.gov where everyone interested can read the details.

In the US, there was a 94% reduction of cervical cancer screenings during the initial phase of the pandemic compared to the same period the previous years, and although the screening has partially recovered since then, the cervical cancer screening rates are still 10% below pre-pandemic levels (*Mast et al., 2022*). Likewise, in England a 43–91% drop per month of received screening samples was observed during the period April to June 2020 and by April 2021 there was still 6.4% fewer samples than expected (*Castanon et al., 2022*). Thus, although there was a prompt re-initiation of screening through re-opening of screening services and catch-up screening during the period following the initial phase of the pandemic, the impact of the disruption was significant. This contrasts with our findings where the disruption was used as an opportunity to advance the program, with lasting improvements already materializing as a greatly improved screening coverage, lower costs of sample taking and increased use of HPV vaccines.

Self-sampling in cervical screening is well known to improve participation among women who seldom or never attended screening (*Sultana et al., 2016*). The increase in population coverage seen is at least in part due to that the sending of self-sampling kits resulted in improved attendance in particular among previously non-attending women.

As the cancellation of non-emergency healthcare also involved canceling of follow-up and treatment of newly detected screen-positive women, it has been speculated that a rise in cervical cancer would result (*Castanon et al., 2022*; *Daniels et al., 2021*). We could, however, not see any such effect in incidence data from the Swedish Cancer Registry (https://www.socialstyrelsen.se/), probably because non-emergency healthcare was allowed again in June 2020 after a less than 3 months disruption.

In summary, the major COVID-19 outbreak necessitated several emergency changes to the cervical screening program. These have resulted in several major and lasting improvements of the cervical cancer prevention strategy that are likely to promote an accelerated elimination of HPV and cervical cancer.

## Acknowledgements

We thank the staff of the regional cancer center Stockholm Gotland for discussions and extraordinary efforts to maintain screening and cervical cancer prevention under difficult circumstances.

We also thank the staff of the Swedish National cervical screening registry for careful data collection and analyses as well as for openly sharing everything on internet.

Finally, we thank Helena Andersson for assistance in preparing the manuscript.

The work of the Swedish National Cervical Screening registry was supported by the Swedish Association of Local Authorities and Regions and the Swedish Cancer Society (20 1199 UsF 02 H). Miriam Elfström and Joakim Dillner are supported by the European Union's Horizon 2020 Research and Innovation Program, RISCC under grant agreement No. 847845. The Swedish even faster campaign is supported by the Swedish government and by the Stockholm county.

The funding agencies have had no role in the design, execution, or interpretation of the study or in the decision to submit for publication.

## Additional information

### Funding

| Funder | Grant reference number | Author |
| --- | --- | --- |
| Swedish Association of Local Authorities and Regions | | Joakim Dillner |
| Cancerfonden | 20 1199 UsF 02 H | Joakim Dillner |

| Funder | Grant reference number | Author |
| --- | --- | --- |
| Horizon 2020 Framework Programme | 847845 | Miriam Elfström Joakim Dillner |
| Swedish Government | | Joakim Dillner |
| Stockholm County | | Joakim Dillner |

The funders had no role in study design, data collection, and interpretation, or the decision to submit the work for publication.

## Author contributions

Miriam Elfström, Conceptualization, Data curation, Formal analysis, Validation, Investigation, Project administration, Writing – review and editing; Penelope Grace Gray, Data curation, Formal analysis, Validation, Writing – review and editing; Joakim Dillner, Conceptualization, Resources, Supervision, Funding acquisition, Investigation, Visualization, Methodology, Writing - original draft, Project administration, Writing – review and editing

## Author ORCIDs

Joakim Dillner (iD) http://orcid.org/0000-0001-8588-6506

## Ethics

The publication of official statistics by the National Cervical Screening Registry is supported by these ethical permissions: Swedish Ethical Review Authority. Ref numbers: 2011/1026-31/4 and 2023-00289-02. The registry linkages using the NKCx were approved by the National Ethical Review Agency of Sweden (decision number 2023-00289-02). The agency is appointed directly by the government of Sweden, chaired by a senior judge and has the authority to determine requirements for consent, was not required for this study.

## Decision letter and Author response

Decision letter https://doi.org/10.7554/eLife.80905.sa1
Author response https://doi.org/10.7554/eLife.80905.sa2

## Additional files

### Supplementary files

• MDAR checklist

### Data availability

The data in this report derive from publicly available data on new Swedish regulations and strategies used during the pandemic (major official websites https://www.socialstyrelsen.se/ and https://www.regeringen.se/). Results on population coverage and number of screening invitations/self-sampling kits sent are derived from the website of the Swedish National Cervical Screening Registry of Sweden (https://www.nkcx.se/).

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
