## [Editor Report]

This is a valuable piece of work given the scope of the intervention(s) and the precedent it sets i.e. a crisis can in fact accelerate positive changes in screening that have been academic possibilities rather than practical realities. The evidence is solid since data were obtained from the national cervical screening registry during the pandemic. The work will be of broad interest to researchers and policy makers involved in cervical cancer screening.

---

## [Decision Letter]

**Decision letter after peer review:**

Thank you for submitting your article "Cervical cancer screening improvements with self- sampling during the COVID-19 pandemic" for consideration by *eLife*. Your article has been reviewed by 3 peer reviewers, including Johannes Berkhof as Reviewing Editor and Reviewer #1, and the evaluation has been overseen by a Senior Editor. The following individual involved in review of your submission has agreed to reveal their identity: Kate Cuschieri (Reviewer #3).

As is customary in *eLife*, the reviewers have discussed their critiques with one another. What follows below is the Reviewing Editor's edited compilation of the essential and ancillary points provided by reviewers in their critiques and in their interaction post-review. Please submit a revised version that addresses these concerns directly. Although we expect that you will address these comments in your response letter, we also need to see the corresponding revision clearly marked in the text of the manuscript. Some of the reviewers' comments may seem to be simple queries or challenges that do not prompt revisions to the text. Please keep in mind, however, that readers may have the same perspective as the reviewers. Therefore, it is essential that you attempt to amend or expand the text to clarify the narrative accordingly.

Essential revisions:

1) This paper has strong points and uses a rich data set but all three reviewers agree that the conclusion about the effect of primary HPV self-sampling on screening coverage, in particular among underscreened women, is too strong. The authors do not convincingly show that coverage under self-sampling is better than before the pandemic nor that self-sampling has led to improved participation in previously non-attending women. Conclusions should be in line with the presented data and all requests for more detailed information about self-sampling coverage data should be carefully addressed by the authors. The paper would benefit from characterizing the population of HPV self-sampling participants, in particular by showing data on the time to previous screen.

2) With regard to the HPV EVEN FASTER concept, this approach is interesting and novel but may be difficult to understand. A few more details on the concept would be helpful for the reader as well as more information on the implementation status of the national campaign (reviewer #1 and #2).

*Reviewer #1 (Recommendations for the authors):*

The data are interesting, but the analysis and presentation are rather crude. The scientific value of the paper in its current form is limited.

1. Data in figure 1 and Table 1 seem to come from different sources, but do not match. Why was the reduction in screening coverage only 4% when there were only a small number of invitations in April 2020 – Dec 2020?

2. The most obvious reason that the screening coverage in 2021 was back at pre-pandemic levels is that the number of invitations was normal again. Please add monthly figures of year 2021 to Figure 1 to illustrate this.

3. In the abstract, the claim is made that a strategy with organized primary self-sampling has resulted in a major improvement of population test-coverage. The increase was only 4% compared to year 2020 (which is not a major improvement), was not observed in the age range 51-60, and levels did not exceed pre-pandemic screening coverage levels. Therefore, it seems fair to conclude that the improvement was achieved because women were re-invited for screening again. Other countries probably showed similar screening coverage patterns without primary self-sampling when there was no concern anymore about crowding (because of availability of vaccines). Please tone down the claim in the abstract.

4. The authors also indicate in the discussion that self-sampling may reach underscreened women. The paper would benefit from showing year 2021 coverage data stratified by time to previous screen.

5. To gain insight into the acceptance of primary self-sampling, please present year 2021 data on the number of women that sent in a self-sampling kit and the number of women indicating that they preferred a provider-collected test.

6. The authors compare their coverage results with other countries (UK and US) where screening rates remained lower than expected. It is questionable whether this has anything to do with primary self-sampling as countries strongly differ in culture, attitude, trust in authorities, health systems etc. It is imaginable that in some countries, the introduction of self-sampling would actually lead to a decrease in screening coverage because of a low trust in the accuracy of a self-collection test. Please elaborate on this in the discussion.

7. A big concern about the Covid19 pandemic is that women who received an invitation in the first year of the pandemic did not show up at all. Were some of these women re-invited for self-sampling in 2021?

8. The authors describe the launch of a concomitant vaccination and screening campaign. No data are presented to show that this campaign was successful. The statement that Maternity Care Clinics expressed interest seems to indicate that this strategy has not yet been implemented and/or endorsed by the health authorities.

*Reviewer #2 (Recommendations for the authors):*

My detailed comments are below.

Methods

Perhaps worth stating the formula used to calculate coverage in the methods for clarity.

Some of the methods mentioned in the abstract are in the results in the main manuscript.

Results

In March 2021 when self-sampling was launched as the primary test was that excluding first attenders, those on follow-up and pregnant? If so worth clarifying as currently not clear.

Further ahead you state the self-sample strategy was implemented in February 2021, is this a discrepancy?

It was not clear whether the strategy to test and vaccinate at the same appointment was implemented or whether there is evidence that maternity clinics would like to do this but it has not been implemented. I think a bit more background on EVEN FASTER is needed in the methods (or result), I didn't realise it was part of a trial till the discussion.

Also perhaps clarify whether you wait for the HPV result prior to vaccinating or whether you vaccinate regardless of the HPV results.

How did laboratories 1 to 3 manage to maintain invitation levels similar to pre-covid but the others did not?

Discussion

There was not enough evidence presented in the manuscript to conclude that "The most likely explanation for the large increase in population coverage seen is that the sending of self-sampling kits resulted in improved attendance in particular among previously non-attending women."

The fact that coverage was very similar to that observed pre-pandemic suggests that the roll-out of self-sampling may have brought in those that were due to attend in 2020 but were either not invited or were not keen to go in person due t COVID. No evidence is presented on previous screening history of people returning a self-sample.

Similarly, no evidence was presented on cervical cancer rate, could the authors at least reference national cancer registration? Also, I would worry that the increase in cancer will come in 2023 onwards among women who missed a whole screening round.

Figure 1 title includes "Laboratory_ID==088" – please delete or add footnote to explain.

*Reviewer #3 (Recommendations for the authors):*

I enjoyed reading this manuscript – some comments are below.

Re "It is well known that when self-samples ……can be used to increase population coverage of screening, with improved attendance among under-screened and hard-to-reach women".

I think this is a slight generalisation/overstatement – the magnitude of increase in coverage v much varies according to geography/setting; there have been some exercises/programmes where offering self sampling vs status quo (ie reminder letter) has been modest at best e.g. [The STRATEGIC trial. J Med Screen. 2018 Jun;25(2):88-98].

Re "Although HPV-testing has been shown to have higher sensitivity in all age groups and higher specificity for women aged 30 or older compared to cytology-based screening." Again, this is a nitpick but historically the specificity of cytology in women over 30 has exceeded that of HPV testing in countries that had national, organised cytology based programmes with investment in EQA, training mandates etc.

I appreciate this is a concise manuscript and that one could revert to (15), but as a standalone piece, I was a little unclear on who exactly was eligible/invited for self sampling- I have mentioned this in the public commentary. I sentence or two of explanation would likely cover it.

---

## [Author Response]

Essential revisions:1) This paper has strong points and uses a rich data set but all three reviewers agree that the conclusion about the effect of primary HPV self-sampling on screening coverage, in particular among underscreened women, is too strong. The authors do not convincingly show that coverage under self-sampling is better than before the pandemic nor that self-sampling has led to improved participation in previously non-attending women. Conclusions should be in line with the presented data and all requests for more detailed information about self-sampling coverage data should be carefully addressed by the authors. The paper would benefit from characterizing the population of HPV self-sampling participants, in particular by showing data on the time to previous screen.

In the revised article we have added exactly this information, by performing individual-level registry linkages.

2) With regard to the HPV EVEN FASTER concept, this approach is interesting and novel but may be difficult to understand. A few more details on the concept would be helpful for the reader as well as more information on the implementation status of the national campaign (reviewer #1 and #2).

This was not a main subject of the paper, merely mentioned as another positive side-effect of the Covid pandemic. It is indeed a novel concept and for a more comprehensive understanding, a whole paper should be dedicated to it.

Reviewer #1 (Recommendations for the authors):The data are interesting, but the analysis and presentation are rather crude. The scientific value of the paper in its current form is limited.1. Data in figure 1 and Table 1 seem to come from different sources, but do not match. Why was the reduction in screening coverage only 4% when there were only a small number of invitations in April 2020 – Dec 2020?

Coverage is calculated over a screening interval. For example, if the screening interval is 5 years and during 1 year no samples are taken at all – the coverage will go down by 20%.

2. The most obvious reason that the screening coverage in 2021 was back at pre-pandemic levels is that the number of invitations was normal again. Please add monthly figures of year 2021 to Figure 1 to illustrate this.

The system was changed from sending invitations to sending out self-sampling kits instead.

3. In the abstract, the claim is made that a strategy with organized primary self-sampling has resulted in a major improvement of population test-coverage. The increase was only 4% compared to year 2020 (which is not a major improvement), was not observed in the age range 51-60, and levels did not exceed pre-pandemic screening coverage levels. Therefore, it seems fair to conclude that the improvement was achieved because women were re-invited for screening again. Other countries probably showed similar screening coverage patterns without primary self-sampling when there was no concern anymore about crowding (because of availability of vaccines). Please tone down the claim in the abstract.

We do not have data on what other countries did and what effect it might have had in other places. We have demonstrated that for us, this intervention did solve the problem.

4. The authors also indicate in the discussion that self-sampling may reach underscreened women. The paper would benefit from showing year 2021 coverage data stratified by time to previous screen.

This has been added to the revised version.

5. To gain insight into the acceptance of primary self-sampling, please present year 2021 data on the number of women that sent in a self-sampling kit and the number of women indicating that they preferred a provider-collected test.

There was no option for women to indicate that they preferred a provided-collected test.

6. The authors compare their coverage results with other countries (UK and US) where screening rates remained lower than expected. It is questionable whether this has anything to do with primary self-sampling as countries strongly differ in culture, attitude, trust in authorities, health systems etc. It is imaginable that in some countries, the introduction of self-sampling would actually lead to a decrease in screening coverage because of a low trust in the accuracy of a self-collection test. Please elaborate on this in the discussion.

Formally, anything is possible. We have shown that when a reasonably large population-based program switched strategy, the strategy switch was successful.

7. A big concern about the Covid19 pandemic is that women who received an invitation in the first year of the pandemic did not show up at all. Were some of these women re-invited for self-sampling in 2021?

If by “invitation” is meant “received a self-sampling kit by mail”, then all women due for screening during 2020 who did not attend did received the self-sampling kit in 2021.

8. The authors describe the launch of a concomitant vaccination and screening campaign. No data are presented to show that this campaign was successful. The statement that Maternity Care Clinics expressed interest seems to indicate that this strategy has not yet been implemented and/or endorsed by the health authorities.

The Even Faster campaign was only mentioned briefly as another positive effect of the Covid epidemic. Going into the details of it is outside the scope of this paper, but I can assure the readers that it is implemented (>100,000 participants so far) and duly endorsed by the health authorities.

Reviewer #2 (Recommendations for the authors):My detailed comments are below.MethodsPerhaps worth stating the formula used to calculate coverage in the methods for clarity.Some of the methods mentioned in the abstract are in the results in the main manuscript.ResultsIn March 2021 when self-sampling was launched as the primary test was that excluding first attenders, those on follow-up and pregnant? If so worth clarifying as currently not clear.

Yes, we have rewritten the wording to make this more clear.

Further ahead you state the self-sample strategy was implemented in February 2021, is this a discrepancy?

The policy switch had some piloting before full implementation.

It was not clear whether the strategy to test and vaccinate at the same appointment was implemented or whether there is evidence that maternity clinics would like to do this but it has not been implemented. I think a bit more background on EVEN FASTER is needed in the methods (or result), I didn't realise it was part of a trial till the discussion.

The Even Faster campaign was only mentioned briefly as another positive effect of the Covid epidemic. Going into the details of it is outside the scope of this paper, but I can assure the readers that it is implemented (>100,000 participants so far) and duly endorsed by the health authorities.

Also perhaps clarify whether you wait for the HPV result prior to vaccinating or whether you vaccinate regardless of the HPV results.

The Even Faster campaign was only mentioned briefly as another positive effect of the Covid epidemic. Going into the details of it is outside the scope of this paper, but in previously published papers on the concept it is clear that vaccination and testing is done at the same time (no waiting for results).

How did laboratories 1 to 3 manage to maintain invitation levels similar to pre-covid but the others did not?

The numbers on the X-axis referred to months of the year. 1,2,3 are January-March when screening went on as usual, before it was stopped. We have revised the Figure legend to make this more clear.

DiscussionThere was not enough evidence presented in the manuscript to conclude that "The most likely explanation for the large increase in population coverage seen is that the sending of self-sampling kits resulted in improved attendance in particular among previously non-attending women."The fact that coverage was very similar to that observed pre-pandemic suggests that the roll-out of self-sampling may have brought in those that were due to attend in 2020 but were either not invited or were not keen to go in person due t COVID. No evidence is presented on previous screening history of people returning a self-sample.

In the revised version, we have added this data (by individual-level registry linkages).

Similarly, no evidence was presented on cervical cancer rate, could the authors at least reference national cancer registration? Also, I would worry that the increase in cancer will come in 2023 onwards among women who missed a whole screening round.

This has been added.

Figure 1 title includes "Laboratory_ID==088" – please delete or add footnote to explain.

Changed as requested.

Reviewer #3 (Recommendations for the authors):I enjoyed reading this manuscript – some comments are below.Re "It is well known that when self-samples ……can be used to increase population coverage of screening, with improved attendance among under-screened and hard-to-reach women".I think this is a slight generalisation/overstatement – the magnitude of increase in coverage v much varies according to geography/setting; there have been some exercises/programmes where offering self sampling vs status quo (ie reminder letter) has been modest at best e.g. [The STRATEGIC trial. J Med Screen. 2018 Jun;25(2):88-98].

The statement was our best assessment of the literature, as also indicated by systematic reviews.

Re "Although HPV-testing has been shown to have higher sensitivity in all age groups and higher specificity for women aged 30 or older compared to cytology-based screening." Again, this is a nitpick but historically the specificity of cytology in women over 30 has exceeded that of HPV testing in countries that had national, organised cytology based programmes with investment in EQA, training mandates etc.

This was how we had interpreted the data. Formal meta-analyses on this point is outside the scope of this paper.

I appreciate this is a concise manuscript and that one could revert to (15), but as a standalone piece, I was a little unclear on who exactly was eligible/invited for self sampling- I have mentioned this in the public commentary. I sentence or two of explanation would likely cover it.

We have tried to be as clear as possible that everyone eligible for screening was instead sent a self-sampling kit. Maybe the question is about the exceptions: that provider-collected sampling of pregnant women in maternity care continued and that the first screens for women entering the program (at age 23-25) was still by invitation to provider collected sampling, We have tried to describe this more clearly now.